# Endophytic Fungi of Tomato and Their Potential Applications for Crop Improvement

**Martina Sinno** [1,*,†], **Marta Ranesi** [1,*,†], **Laura Gioia** [1], **Giada d'Errico** [1] and **Sheridan Lois Woo** [2,3,4]

1   Department of Agricultural Sciences, University of Naples Federico II, 80055 Portici (NA), Italy; laura.gioia@unina.it (L.G.); giada.derrico@unina.it (G.d.)
2   Department of Pharmacy, University of Naples Federico II, 80131 Napoli, Italy; woo@unina.it
3   Task Force on Microbiome Studies, University of Naples Federico II, 80131 Naples, Italy
4   National Research Council, Institute for Sustainable Plant Protection, 80055 Portici, Italy
*   Correspondence: martina.sinno@unina.it (M.S.); marta.ranesi@unina.it (M.R.); Tel.: +39-340-928-4138 (M.S.); +39-329-1461-266 (M.R.)
†   These authors contributed equally to this work.

**Abstract:** Endophytic fungi (EF) are increasingly gaining attention due to the numerous benefits many species can offer to the plant host, while reducing the application of chemicals in agriculture, thus providing advantages to human health and the environment. The growing demand for safer agrifood products and the challenge of increasing food production with a lower use of pesticides and fertilizers stimulates investigations on the use and understanding of EF. Other than direct consequences on the plant damaging agents, these microorganisms can also deliver bioactive metabolites with antimicrobial, insecticidal, or plant biostimulant activities. In tomato, EF are artificially introduced as biological control agents or naturally acquired from the surrounding environment. To date, the applications of EF to tomato has been generally limited to a restricted group of beneficial fungi. In this work, considerations are made to the effects and methods of introduction and detection of EF on tomato plants, consolidating in a review the main findings that regard pest and pathogen control, and improvement of plant performance. Moreover, a survey was undertaken of the naturally occurring constitutive endophytes present in this horticultural crop, with the aim to evaluate the potential role in the selection of new beneficial EF useful for tomato crop improvement.

**Keywords:** endophytes; biocontrol; biostimulants; induced systemic resistance; ISR; plant pathogens; fungal entomopathogens

## 1. Introduction

Different approaches can be used to discover alternatives to chemical pesticides, to prevent or control harmful plant biotic agents. In recent years, the interest in the biological control of plant pests and pathogens has surged to meet the requirements for more environmentally friendly options to synthetic chemicals. Consequently, the method for the use of microbial biological control agents (mBCAs), as natural antagonists to suppress herbivores and organisms that cause disease, has increased and improved. Fungi are among the most important mBCAs selected for this application due to their ease of isolation, selection from a vast number of known non-pathogenic strains, morphological structures for conservation and delivery, adaptation to numerous engineering fermentation technologies in industry, manageability in formulations, as well as their capability to secrete and over-express endogenous proteins or nontoxic exogenous compounds [1]. Furthermore, many beneficial fungi are

known to promote plant growth and act as plant biostimulants or biofertilizers, thus their application in agriculture may also reduce the use of chemical fertilizers [2].

Among the plant favorable fungi, fungal endophytes, in particular, have been gaining increased attention because of the numerous benefits they can offer directly to the plant host with the intimate interaction established during the colonization of the plant tissues [3–7]. Since their discovery, endophytes have been isolated from different vegetative structures, many diverse plant species, both in natural uncultivated, as well as in agricultural environments [8]. These endophytic fungi (EF) represent a microbial community with an enormous reserve of biodiversity, originating from diverse ecological niches and host tissues ranging from the algae living in marine environments [9] to trees in the forest ecosystems [10].

These microorganisms have the ability to colonize plants without causing any symptoms [11], establishing a plant-fungi association inside the living plant tissue, that may occur within roots, stems, and/or leaves, and they emerge from the plant tissue only at the time of sporulation or upon senescence of the host [12–14]. The fossil record indicates that plants have had associations with endophytic [15] and mycorrhizal [16] fungi for more than 400 million years, a relationship that has likely existed since the time when plants first colonized land, thus playing a long and important role in the driving force of evolution, and life on land. In recent decades, scientific evidence has demonstrated that non-pathogenic microbes, endo- or exo-inhabitants of plants, may be associated with latent pathogens or early colonizing saprophytes that could actively grow in the living vegetative tissues only at the moment that plant defense responses waned or the plant initiated the phase of senescence [17]. The specific interaction between the host plant and its microbial partners ranges on a continuum from neutralism towards mutualism and antagonism, in which the nature of the relationship may change during the lifecycle of the plant depending upon environmental as well as intrinsic factors [18–20].

Currently, the concept of the plant microbiome considers a viewpoint on plant-microbe evolution, in which the plant and the microbiota have evolved together, with the microorganisms providing advantages and versatility to the plant in its ecosystem: an exchange of the plant as a habitat and source of nutrition, with some endophytic microorganisms producing benefits to the host plant that include the stimulation of growth and development, adaptation to the environment and abiotic stress tolerance [21,22]. More recently, it has been reported that EF can also have a protective role against attack by insects [23], pathogens [24], and nematodes [25], thus acting as multiple plant defenders or biocontrol agents.

Like most of the beneficial fungi, EF are known to secrete a vast number of bioactive secondary metabolites that are primarily responsible for the observed useful effects since they can stimulate the plant defense response and growth, as well as exert a direct antimicrobial or insecticidal effect [2,26–28]. Indeed, the remarkable advantage of these microorganisms is due to the in-depth relationship within the plant host that allows the immediate availability of the secreted active molecules within plant tissues [28,29]. In a broad sense, EF are producers of bioactive metabolites for which the plant constitutes a delivery system; in the case of insecticidal or antimicrobial molecules, the plant serves as a pipeline for the translocation of these compounds to the target pathogen or pest, thus EF act as a biopesticide [29,30].

Tomato belongs to the Solanaceae family, and it is one of the most commonly cultivated vegetable crops worldwide. *Solanum* section *Lycopersicon* includes the cultivated tomato (*Solanum lycopersicum* L.) and 12 additional wild relatives, but *S. lycopersicum* is the only domesticated species [31]. The tomato originated from South America, then spread globally through different levels of domestication, starting prior to the 15th century and continuing into Europe, arriving to its present status as one of the most highly consumed food crops of international acclaim. In 2018, the global tomato production reached more than 180 million tons, in which the cultivated area worldwide of harvested tomatoes accounted for almost 5 million ha (FAOSTAT 2018). Due to its extensive global distribution and consumption, tomato is one of the most important horticultural crops farmed, and during its cultivation, it is constantly threatened by pests and pathogens. In recent times, the use of chemical pesticides is

becoming largely restricted in agriculture both due to the negative impact on human health and the environment (EU Directive 2009/128/CE), plus the risks of resistance development in pathogen/pest populations by their use.

The growing demand by consumers for safer food products plus the urgent challenge of increasing food production with lower inputs of pesticides and fertilizers stimulates both the utilization of EF as non-chemical plant beneficials, as well as the investigations that provide further insights into the interactions of these microorganisms with the crop host and the organisms that damage it. Recent studies have focused on the use of EF in protecting and improving tomato crop as an alternative to the chemical approach for crop protection (references in this review). The presence of EF in tomato can be prevalent in the plant if the fungi were introduced as mBCAs, or if the EF were naturally acquired from the surrounding environment and horizontally and/or vertically transmitted into the plant [32]. According to the increasing interest in this argument, the present paper offers a review of the principle EF that have been introduced to tomato for the biological control of various pests and pathogens, plus those applied to improve the plant performance, growth/yield, and quality. Furthermore, a section of this review is dedicated to constitutive EF isolated and identified from tomato, which can serve as a valuable source of new microbial beneficial applications with unexplored potential to improve the production of tomato and other crops.

## 2. Beneficial Effects of EF Introduction on Crops

The beneficial effects executed by EF include pest and pathogen control, plant growth promotion (PGP), improvement of the plant nutrient availability and uptake, and the increased tolerance to abiotic stress, hereby referred to as plant physiology improvement (PPI). EF have been confirmed to affect insect pests feeding on the plants that they colonize [23,33–35], and results in the literature indicated that these microorganisms provided protection from significant herbivory damage they cause to crops. Insect pests have been noted to be affected by EF in numerous ways such as reduction of developmental rate [36,37], deterrence of feeding on the colonized plant [38,39], retarded insect growth, higher mortality, and lower oviposition [40,41]. One of the hypothesized mechanisms underlying these effects is the bioaccumulation of secondary metabolites and mycotoxins produced by the EF within the plant tissues [34]. Moreover, these microorganisms are known to defend their plant host from pathogens attack [3–7]. This biocontrol action could be exerted through direct mechanisms including food and space competition, parasitism, and antibiosis [39,42–44]. While colonizing host plants, EF stimulate the attacked plant to create a barrier (biochemical or mechanical) that inhibits pathogenic organisms from penetrating the same tissue hence preventing the occurrence of diseases [45]. An important indirect mechanism involved in plant protection is the induction of plant resistance which is implemented by the alteration of the biochemical signaling pathways of the plant that modulate the resistance-related genes which are triggered by the endophytic colonization [46–51]. PGP implemented by EF is characterized by the improvement of the above- and/or below-ground biomass [52–54] while PPI effects include the increase of nutrients uptake, particularly nitrogen and phosphorus [55,56], and enhanced tolerance to abiotic stress including drought, salt, and heat [1]. EF can be, thus, utilized as biofertilizers as they improve the nutrient uptake firstly enhancing the plant root system and secondly, in the case of entomopathogenic endophytes, reallocating the insect-derived nitrogen to the host plant. In fact, EF, after feeding on insects in the rhizosphere, may translocate the adsorbed nitrogen to the host plant towards the association with the root system [55].

## 3. Introduced Endophytes of Tomato

The utilization of EF as biological control agents (BCA) represents a potential alternative that meets the growing need for more eco-sustainable agriculture. According to this new perspective, in recent years many studies have been performed, introducing EF on tomato to test their effects on plant performance. Among these introduced EF, a consistent number of species belongs to a group of fungi classified as entomopathogens, fungi that are pathogens to insects, many isolated from asymptomatic plants,

including *Akanthomyces* spp., *Beauveria bassiana*, *Clonostachys rosea*, *Cordyceps farinosa* (formerly *Isaria farinosa*), *Lecanicillium* spp., and *Sarocladium* spp. (formerly *Acremonium* spp.) [39,57–61]. The natural occurrence of these fungi within the plant tissues suggests their ability to endophytically colonize a wide range of plants.

To date, in tomato as for many other horticultural crops, the artificial introduction of EF has been limited to a restricted group of beneficial microbes which include species belonging to the genus *Sarocladium*, *Beauveria*, *Metarhizium*, *Fusarium*, *Penicillium*, *Serendipita* (formerly *Piriformospora*), *Pochonia*, *Pythium*, and *Trichoderma* [36,37,62–68]. Furthermore, some introduced species belong to the Dark Septate Endophytes (DSE) such as *Neocosmospora haematococca* (formerly *Nectria haematococca)* and *Periconia* spp. DSE represent a large group of root-inhabiting endophytes not yet well defined taxonomically and/or ecologically that are distinguished as a functional group based on the presence of darkly melanized septa. DSE are ubiquitous and abundant in various ecosystems and playing an interesting role in contrasting pathogens as they can improve plant tolerance to abiotic stress [69], growth [70], and nutrient uptake [71]. In short, DSE may play an important role in the ecophysiology of plants. However, almost a century after their discovery, little is still known about the role of these mysterious and rather elusive fungal symbionts.

### 3.1. Biocontrol

The biocontrol of pests and pathogens has been the most documented beneficial effect explicated by the artificial introduction of endophytic fungi to tomato. In this context, 41 scientific papers report the use of mBCAs that focuses on crop protection and the consequences on the organisms that are deleterious to tomato (Table 1).

In particular, concerning the insect pests, biocontrol potential of EF was evidenced for the negative effects on *Aphis gossypii* [72], *Bemisia tabaci* [73,74], *Chortoicetes terminifera* [72], *Helicoperva armigera* [75,76], *Nesidiocoris tenuis* [77], *Spodoptera exigua* [78], *S. littoralis* [23], *Tuta absoluta* [79,80], and *Trialeurodes vaporariorum* [43]. Furthermore, *Neocosmospora solani* (formerly *Fusarium solani*) increased tomato defense against infestations of the red spider mite, *Tertranychus urticae* [46], while several endophytic species were able to induce resistance to the root-knot nematode *Meloidogyne incognita* [25,81–83]. The overall effects observed on tomato pests included: increased mortality, feeding deterrence, reduced growth rate and reproduction, reduced infestation, egg masses colonization, and increased plant defense.

Regarding disease control, EF were reported to counteract the infection of the bacterial pathogen *Clavibacter michiganensis* subsp. *michiganensis* [84], fungal pathogens *Fusarium oxysporum* f. sp. *lycopersici* [84–92], *Rhizoctonia solani* [93] and *Botrytis cinerea* [42,94]. In the above-mentioned papers, the reported effects of endophytic colonization on pathogen control were noted with reduced disease symptoms and a disease-suppressive effect.

In most papers, the authors suggested that the reduced impact of pests and diseases was due to plant resistance induced by its microbial partner. The mechanism underlying this induced resistance are subdivided into two main categories: Systemic Acquired Resistance (SAR) and Induced Systemic Resistance (ISR). SAR is induced by the plant local infection by latent pathogens and is effective against a broad range of harmful plant biotic agents, it is mediated by salicylic acid (SA) and associated with pathogenesis-related proteins [95]. This is the case of *N. solani* strain Fs-K which was reported to induce plant resistance against *Septoria lycopersici* through a SAR mechanism [87]. On the other hand, ISR is triggered by the endophytic colonization of beneficial microorganism such as plant growth-promoting rhizobacteria and EF that involves a priming process of the plant which results in more efficient activation of its defense responses against pests and pathogens [96]. It is mediated by jasmonic acid (JA) and ethylene (ET) [97]. This is the case of *Trichoderma hamatum* which is reported to induce resistance against the tomato bacterial spot caused by *Xanthomonas euvesicatoria* [98]. Nonetheless, SAR and ISR may be two distinct but overlapping mechanisms as a result of crosstalk of the two hormonal pathways [97], as noted for *Trichoderma harzianum* which induced plant resistance against *M. incognita* through priming plant defense with both SA and JA stimulation [99].

### 3.2. Plant Growth Promotion and Plant Physiology Improvement

Plant growth promotion of tomato attributed to endophytic colonization has been well-documented (Table 1). Thirteen studies have indicated that there was evident PGP as demonstrated by the improvement of the root system with greater root length, biomass, and dry weight [69,83,88,94,100–102], increased plant height, shoot biomass, and fresh or dry weight [69,70,83,88,89,100,101], plus enhanced plant production with anticipated flowering and fruiting times, and increased fruit weight [102].

Moreover, a few articles reported improvement of the plant nutrient uptake and the increased tolerance to abiotic stress (PPI) (Table 1). Improved plant uptake of iron (Fe) [103], organic nitrogen (N) [70,71], and inorganic potassium (K) [70], have been demonstrated to be a consequence of the plant endophytic colonization with some fungal species namely *B. bassiana*, *Periconia macrospinosa* (DSE) plus an unidentified species also belonging to DSE. Additionally, some studies have highlighted that the presence of the endophyte confers tolerance to diverse abiotic stress such as drought [69], salinity [104], and metals [101].

**Table 1.** Effects of introduced fungal endophytes on tomato plants in terms of Plant Growth Promotion (PGP) and Plant Physiology Improvement (PPI), and Biocontrol (BC) of pest and pathogens.

| Fungal Species | Effects | |
| --- | --- | --- |
| | PGP and PPI | BC |
| *Sarocladium strictum* * | Increased number of xylem vessels within the shoots [84] | Increased mortality of larvae of *Trialeurodes vaporariorum* [105] |
| *Sarocladium kiliense* * | | Reduced symptoms caused by *Fusarium oxysporum* f. sp. *lycopersici* and *Clavibacter michiganensis* subsp. *michiganensis* [84] |
| *Beauveria bassiana* | Enhanced terpene production [78] Improved iron (Fe) nutrition [103] | ISR vs. *Rhizoctonia solani* [93] ISR vs. *Botrytis cinerea* [42] ISR vs. *F. oxysporum* f. sp. *lycopersici* [85] Increased mortality of *Tuta absoluta* [79,80] Reduced incidence of *Fusarium oxysporum* f. sp. *lycopersici* and *Helicoverpa armigera* [106] Increased mortality of *Helicoperva armigera* [75,106] Increased mortality of *Bemisia tabaci* [73] Feeding deterrent for *Bemisia tabaci* [74] Increased mortality of *Spodoptera littoralis* [23] Reduced growth rate of *Spodoptera exigua* [78] Reduced reproduction of *Aphis gossypii* and reduced growth rate of *Chortoicetes terminifera* [72] |
| *Metarhizium anisopliae* | Increased plant height, root length, shoot and root dry weight [100] | Increased mortality of *Spodoptera littoralis* [23] |
| *Fusarium oxysporum* | | ISR vs. *F. oxysporum* f. sp. *lycopersici* [86] ISR vs. *Meloidogyne incognita* [81,82] Fermentation broth with anti-oomycete activity vs. *Pythium ultimum*, *Phytophthora infestans* and *Phytophthora capsici* [24] Reduced infestation of *Trialeurodes vaporariorum* [42] ISR vs. *Nesidiocoris tenuis* [77] |
| *Neocosmospora solani* * | | ISR vs. *F. oxysporum* f.sp. *radicis-lycopersici* [87] SAR vs. *Septoria lycopersici* [87] Increased tomato defenses against *Tertranychus urticae* [107] |
| *Fusarium* spp. | Increased roots length, shoots height and plant fresh weight [88] | ISR vs. *Fusarium oxysporum* f. sp. *radicis-lycopersici* [88] |
| *Neocosmospora haematococca* * (DSE) | Drought stress tolerance, improved plant growth, and proline accumulation [69] | |
| Unidentified (DSE) | Increased aboveground plant dry biomass and increased uptake of organic N and inorganic K [70] Salinity stress tolerance [104] | |
| *Penicillium simplicissimum* * | Metal stress tolerance [101] Increased shoot length and biomass under normal and Al stress conditions [101] | |
| *Periconia macrospinosa* (DSE) | Improved organic N uptake and plant biomass when organic nutrients are present [71] | |
| *Serendipita indica* * | Increased fresh weight [89] Accelerated vegetative and generative development [108] | ISR vs. Tomato yellow leaf curl virus [109] Disease-suppressive effect vs. *Verticillium dahliae* and *F. oxysporum* [89–91] Reduced infestation of *Meloidogyne incognita* [25] |
| *Pochonia chlamydosporia* | Increased root and shoot growth [83] Anticipated flowering and fruiting times, increased fruit weight and root growth [102] | Colonizes egg masses of *Meloidogyne incognita* [83] |
| *Pythium oligandrum* | | ISR vs. *Ralstonia solanacearum* [110] ISR vs. *Fusarium oxysporum* f. sp. *lycopersici* [92] ISR vs. *B. cinerea* [111] |
| *Trichoderma atroviride* | Increased root and shoot growth depending on the tomato cv [94] | Reduced infestation of *Trialeurodes vaporariorum* [43] ISR vs. *Botrytis cinerea* [94] |
| *Trichoderma hamatum* | | ISR vs. *Xanthomonas euvesicatoria* (tomato bacterial spot) [98] ISR and SAR vs. *Meloidogyne incognita* [99] |
| *Trichoderma harzianum* | Increased root and shoot growth depending on the tomato cv [94] | ISR vs. *Botrytis cinerea* [94] Reduced desease symptoms caused by *Alternaria solani* and *Phytophtora infestans* [112] |

* scientific names are different from those present in the articles cited due to taxonomic updates to the name presently use.

### 3.3. Methods of Introduction and Detection

In the last decades, it has been demonstrated that several beneficial EF can be artificially introduced on tomato using different inoculation methods and numerous protocols have been developed to successfully achieve this colonization, as well as to detect the fungi within the plant tissues. The methods used for the introduction and detection of EF in tomato plants are summarized in Table 2. The inoculation of EF to tomato plants is mainly achieved with conidial suspension applied by seed treatments, root dipping, soil watering, stem injection, and leaf spraying. Alternatively, the application can be performed by mixing fungal biomass with the transplanting soil. Among the different inoculation techniques, the soil applications, mainly by watering with a conidia suspension, was the most commonly and successfully used technique applied in 18 studies. This was followed by the treatment of seed, as adopted in 15 studies, which was performed by various methods including seed soaking, seed coating, and seed dressing. Seed soaking consisted of placing the tomato seeds in a liquid conidial suspension for 2 to 24 h, before planting. Seed coating involved immerging the seeds in a conidial suspension, stirring them every 30 min for 2–3 h, to cover and adhere the spores to the seed surface, then successively air-drying under sterile conditions [93,94,113]. Seed dressing, was the less common technique, preparing and mixing the seeds in a conidia suspension with continuous shaking for several hours [74,103]. The conidia suspension usually contained a "sticker" such as Tween 80 (0.1–0.01% *v/v*) or methylcellulose (5–10% *v/v*), to ensure a more efficient adhesion of the conidia to the seed surface. Root dipping was another technique commonly used that consisted of dipping the seedling roots in a conidial or propagule suspension for 6 to 24 h prior to transplant [71,76,79,89,106,110].

It is evident that the methods of application were numerous, and the selection of the most efficient method is highly dependent on the specific EF that is employed. The majority of the studies, involving the artificial introduction of EF to tomato, were conducted in a controlled environment, usually with sterilized soil or transplanting substrate, and not in the open field, in order to facilitate the monitoring of the plant colonization. The field application of EF is a challenge that needs to take into account the enormous variability of the environment that could negatively affect the efficacy of the above-mentioned protocols for the introduction. Another critical issue is represented by the transient nature of some endophytes in plant colonization, which explains why, in most cases, the studies do not report details on the time duration of the endophytic colonization. A study by Resquin-Romero [23] indicated that the endophytic colonization of the plant was transient and that the EF–plant interaction was lost after a certain period of time after inoculation. Due to the transient nature of the endophytic colonization, as has been documented in other crops, it is recommended that parallel time-course studies should be performed to monitor the extent of endophytic development, for example, with molecular, microscopy, and/or in vivo re-isolation techniques [68,114–119]. This attests to the difficulty of establishing stable and lasting interactions between the chosen endophyte and its plant host, in the attempt to obtain the potential desired effect.

To ensure that the inoculation of the fungal species is followed by actual endophytic colonization of the plant, it is mandatory to include an experimental stage to detect the EF within the plant. Out of the 52 papers reviewed, 16 studies did not include an endophyte detection assay in their experimental workflow. It is recommended that an analysis of the EF presence should always be included in the study in order to assess the success of the endophytic colonization of the plant, plus monitor the rate of colonization. Moreover, this detection stage should follow an accurate surface-sterilization of the plant tissues to avoid the inadvertent isolation of epiphytic rather than endophytic fungi.

The methods to determine the presence of EF can be divided into three main types: the re-isolation of the EF from the plant tissue, the molecular detection by polymerase chain reaction (PCR), and morphological observation using microscopy techniques. Each method requires the sterilization of the plant material to eliminate the epiphytic microbial community, usually obtained by dipping the tissue in a diluted bleach solution for 1–3 min, that can be followed or proceeded with a brief 70% ethanol bath, completed with rinsing it at least three-times with sterile water. As a check of the efficacy

of the sterilization procedure, aliquots of the rinsed water are also plated, and if bacterial or fungal growth occurs the sample is discarded.

The re-isolation of the fungal colony from the host plant tissue is the most used method to assess the endophytic colonization and is reported in 17 studies. Usually, it follows this protocol: collect, wash, and sterilize the plant material, dissect the vegetal tissues in 1 cm pieces under sterile conditions, and place the pieces on Petri dishes containing solid culture substrate. Most of the authors utilized potato dextrose agar (PDA), supplemented with antibiotics to avoid bacterial contaminations, while others used selective media for the specific EF they were interested in re-isolating [23,80,100,120]. Molecular analysis was also widely used for the identification of the EF and is reported in 12 studies. It was based on the extraction of the DNA from the pre-sterilized plant tissue, and the subsequent amplification by PCR and sequencing of amplicons for specific fungal molecular markers such as the Internal Transcribed Spacer (ITS1 and ITS2) region and the translation elongation factor (TEF). Five manuscripts included the quantitative detection of the EF within the plant tissue using a real-time PCR [77,86,87,107,121].

Eleven studies used microscopy techniques to visually examine the fungal presence within the plant tissues. These techniques included light optical microscopy using stained plant tissues, usually with trypan blue or methyl blue, scanning electron microscope (SEM), and transmission electron microscope (TEM). The microscope analysis was particularly valuable for observing and understanding the EF growth distribution patterns and translocation within the plant tissues, thus providing important information and a deeper insight of the EF colonization that was not possible in comparison to the other methods with the re-isolation or molecular detection.

**Table 2.** Methods of introduction and detection of fungal endophytes in tomato plant with relative cultivar.

| Fungal Species | Tomato Cultivar | Method of EF Inoculation | Detection Method | Location of EF in Plant Tissues | Ref. |
|---|---|---|---|---|---|
| *Sarocladium kiliense* * | Haubner's Vollendung | Fungal biomass mixed with transplanting soil | | Roots | [84] |
| *S. strictum* * | Haubner's Vollendung | Soil watering | Re-isolation from the plant tissue on PDA | Roots | [105] |
| *S. strictum* * | Suso RZÒ F1 hybrid | Soil watering | Re-isolation from the plant tissue on MEA | Roots | [122] |
| *Beauveria bassiana* | Platense | Seed soaking Leaf spraying Root dipping | Re-isolation from the plant tissue on PDA | Leaves | [79] |
| *B. bassiana* | Mobil | Seed coating | Re-isolation from the plant tissue on PDA | | [93] |
| *B. bassiana* | Limachino—INIA | Fungal biomass mixed with transplanting substrate | Re-isolation from the plant tissue on Noble agar | Roots Stem Leaves | [42] |
| *B. bassiana* | Rio Fuego | Soil watering Leaf spraying Stem injection | | | [85] |
| *B. bassiana* | Ace, Early Pack, Money Maker, Peto 86, Prichard, Pusa Ruby, Strain B and LA1478 | Leaf spraying Stem injection | PCR | Stem | [73] |
| *B. bassiana* | Grosse lisse | Leaf spraying | Re-isolation from the plant tissue on PDA | Leaves | [34] |
| *B. bassiana* | Harzfeuer F1 | Leaf spraying | Re-isolation from the plant tissue on selective media | Leaves | [80] |
| *B. bassiana* | Regina | Conidial suspension on wounded rachis | Re-isolation from the plant tissue on selective media | Roots | [120] |

**Table 2.** *Cont.*

| Fungal Species | Tomato Cultivar | Method of EF Inoculation | Detection Method | Location of EF in Plant Tissues | Ref. |
|---|---|---|---|---|---|
| *B. bassiana* | Cal-J, Kilele F1, Anna F1 | Seed soaking | Re-isolation from the plant tissue on SDA | Roots Stem Leaves | [123] |
| *B. bassiana* | Cal-J, Kilele, Anna | Seed soaking | Re-isolation from the plant tissue on SDA | Roots Stem Leaves | [124] |
| *B. bassiana* | Mountain Spring | Seed coating | | | [113] |
| *B. bassiana* | PKM1 | Seed soaking Root dipping | | | [76] |
| *B. bassiana* | PKM1 | Soil watering Seed soaking Root dipping | | | [106] |
| *B. bassiana* | surahi | Root dipping Stem injection Soil inoculum Leaf spray | Re-isolation from the plant tissue on PDA | Leaves | [75] |
| *B. bassiana* | Tres Cantos | Leaf spray | Re-isolation from the plant tissue on selective media | Stem Leaves | [23] |
| *B. bassiana* | Marmande-Cuarenteno | Seed soaking | Re-isolation from the plant tissue on SDCA | Stem Leaves Roots | [35] |
| *B. bassiana* | Castlemart | Seed coating | PCR | Shoot | [78] |
| *B. bassiana* | Hezuo 903 | Leaf spray Root irrigation Reed dressing | PCR | Shoot | [74] |
| *Fusarium* spp. | Rio Grande | Soil watering | PCR | Root Stem | [88] |
| *F. oxysporum* | Montfavet 63-5 | Root application | Real-Time qPCR | Roots Cotyledons | [86] |
| *F. oxysporum* | Furore | Soil application | | Roots | [81] |
| *F. oxysporum* | Moneymaker | Soil watering | | Roots | [82] |
| *F. oxysporum* | Hellfrucht/JW Frühstamm | Soil watering | | Roots | [43] |
| *Neocosmospora solani* * | Pearson | Soil watering | Real-Time qPCR | Roots | [77] |
| *N. solani* * | Ace 55 | Soil watering | Real-Time qPCR | Roots | [107] |
| *N. solani* * | Ace 55 | Soil watering | Microscopy Real-Time qPCR | Roots | [87] |
| *Metarhizium anisopliae* | Hybrid var. 8625 | Soil watering | Re-isolation from the plant tissue on selective media | Roots Shoots Leaves | [100] |
| *M. anisopliae* | Tres Cantos | Leaf spray | Re-isolation from the plant tissue on selective media | Stem Leaves | [23] |
| *M. brunneum* | Ruthje | Encapsulated mycelial biomass | Light microscopy Real-Time qPCR | Stem | [121] |
| *Neocosmospora haematococca* * (DSE) | CO-2 | Soil application of mycelial biomass formulation | Light microscopy | Roots | [69] |
| Unidentified (DSE) | Santa Clara I-5300 | Soil application of mycelial biomass | Light microscopy | Roots | [70] |
| *Penicillium semplicissimum* * | LA2710 | Soil application of mycelia and culture filtrate | | Roots | [101] |
| *Periconia macrospinosa* (DSE) | Hildares F1 | Root dipping in propagule suspension | Light microscopy | Roots | [71] |
| *Serendipita indica* * | Hildares | Root dipping | Re-isolation from the plant tissue on PDA | Roots | [89] |
| *S. indica* * | T07-4, T07-1 | Transplanting substrate application of mycelia | Light microscopy | Roots | [109] |
| *S. indica* * | Nutech | Seed coating (bioformulation) | | Roots | [90] |
| *S. indica* * | Vellayani Vijay | Transplanting substrate application of mycelia | Light microscopy | Roots | [25] |

**Table 2.** *Cont.*

| Fungal Species | Tomato Cultivar | Method of EF Inoculation | Detection Method | Location of EF in Plant Tissues | Ref. |
|---|---|---|---|---|---|
| *Pochonia chlamydosporia* | Durinta | Plating of seedlings on fungal plate cultures | laser-scanning confocal microscopy PCR | Roots | [83] |
| *P. chlamydosporia* | Marglobe | Seed germination on fungal plate cultures | Re-isolation from the plant tissue on CMA PCR | Roots | [102] |
| *Pythium oligandrum* | Micro-Tom | Root dipping | laser scanning microscopy | Roots | [110] |
| *P. oligandrum* | Prisca | Mycelial plugs in proximity of the top root | SEM TEM | Roots | [92] |
| *P. oligandrum* | Prisca | Soil watering | TEM | Roots | [111] |
| *Tricoderma atroviride* | Hellfrucht/JW Frühstamm | Soil application | | Roots | [43] |
| *T. atroviride* | Corbarino, M82, SM36, TA209 | Seed coating | | Roots | [94] |
| *T. hamatum* | Ohio 8245 | Soil application | | Roots | [98] |
| *T. harzianum* | Corbarino, M82, SM36, TA209 | Seed coating | | Roots | [94] |
| *T. harzianum* | Moneymaker | Soil application | | Roots | [99] |
| *T. harzianum* | Arka vikas | Soil watering | | Roots | [112] |

* scientific names are different from those present in the articles cited due to taxonomic updates to the name presently use.

## 4. Constitutive Endophytes of Tomato

EF have been reported to have a crucial role in inducing plant host tolerance to stressful conditions [59], plant defense [32], and plant growth and development [125]. In all-natural or agricultural ecosystems, every plant is colonized by a diversity of soil-borne microorganisms as root endophytes, mycorrhizal fungi, and plant growth-promoting rhizobacteria. Moreover, the analysis of plant–endophyte associations in high-abiotic stress habitats revealed that at least some fungal endophytes confer habitat-specific stress tolerance to the host plants. Without the presence of the habitat-adapted fungal endophytes, these plants were unable to survive in their native habitats [126]. Thus, the naturally occurring EF constitute a poorly exploited resource, rich in terms of biodiversity, representing a pool of potentially beneficial fungi from which the selection of new strains may be obtained for useful applications in agriculture.

Seven studies focused on the naturally occurring EF of tomato and the data are summarized in Table 3. The constitutive EF were comprised of 24 different genera, among which the most represented are *Trichoderma* and *Fusarium*, which included 35 different fungal species. It is interesting to note that some of the fungi reported in Table 3 are commonly recognized as plant beneficial fungi, such as *Trichoderma* spp., *N. solani*, and *Sarocladium implicatum* (formerly *Acremonium implicatum*), while other species are known as plant pathogens, for example, *Alternaria solani*, *Stemphilyum lycopersici* and *Albifimbria verrucaria*. *A. solani* causes early blight of tomato, one of the common foliar diseases of tomato [127], *S. lycopersici* is the causal agent of leaf spot disease on pepino (*Solanum muricatum*) [128], and *A. verrucaria* produces small brown to black spots symptoms on the colonized leaves and stems [129]. Moreover, *A. verrucaria* is also known to be the responsible agent of mycotic keratitis, one of the major causes of ophthalmic morbidity and visual loss globally [130]. This highlights the importance of identifying EF to study their prospective utilization in agriculture, but also to understand the possible implications on human health.

An example of EF use for tomato improvement is provided by the work of Bogner and colleagues [32] that was conducted in five different counties of Kenya with the aim of identifying and characterizing the culturable endophytic mycobiota in the roots of tomato and screening different fungal endophytes for their biocontrol potential towards the root-knot nematode *Meloidogyne incognita*. A total of 76 fungal isolates were obtained, among which the most prevalent species associated with tomato roots were members of the *F. oxysporum* and *N. solani* species complexes. Bioassays

demonstrated the ability of selected non-pathogenic fungal isolates to successfully reduce nematode penetration and subsequent galling, as well as decrease the reproduction capacity of the root-knot nematode *M. incognita*. Most isolates in the *Trichoderma asperellum* and *F. oxysporum* complex were able to reduce the root-knot nematode egg densities by 35–46% in comparison to the treatments with the nonfungal control and the other fungal isolates. Moreover, Tian and colleagues isolated an endophytic fungus from tomato root galls infected with *M. incognita* that was identified as *S. implicatum* based on morphological and molecular identification [131]. The biocontrol potential of *S. implicatum* culture filtrates was tested with the plant and nematodes in vitro, in pot and field experiments. Results from the in vitro test indicated that 96% of second-stage juveniles of *M. incognita* were killed after 48 h. The fungal compounds were also able to suppress egg hatching, the formation of root galls, and reduce the nematode population in the soil.

These findings suggest that naturally occurring EF populations in the soil represent an underestimated and valuable source of microbial diversity with positive impacts on sustainable agricultural production, due to the possibility to reduce the use of chemical products, thus benefiting the environment and human health. Moreover, this highlights the importance of promoting the constitutive endophytic populations in the soil in order to obtain the effective threshold level for biological control of organisms that compromise plant health. Many studies have demonstrated that soil type and plant genotype are the two main variables that affect the establishment of fungal species in the soil community [132–134]. The cultivation system can also influence the microbial species in the soil, whereby fungal abundance was significantly higher in organically farmed fields than the populations found in conventionally farmed that used chemicals [32,132]. In order to successfully develop applications of plant-associated EF in sustainable agricultural production, further investigations are necessary to understand the mechanisms of action and the processes employed by the fungi to produce the beneficial effects, as well as to determine how they can be efficiently utilized in actual practices.

**Table 3.** Naturally occurring constitutive endophytes of tomato.

| Fungal Species | Tomato Cultivar | Main Results | Location of EF in Plant Tissues | Country | Ref. |
|---|---|---|---|---|---|
| *Alternaria solani* *Aspergillus sclerotiorum* *Cochliobolus geniculatus* *Curvularia lunata* * *Fusarium nygamai* *Fusarium sp.* *Fusarium verticillioides* *Stemphylium lycopersici* *Trichoderma asperellum* *Trichoderma lixii* * | Moneymaker | Biological control to the rootknot nematode *Meloidogyne incognita* | Root | Kenya | [32] |
| *Fusarium* spp. | Heinz 9907 Gem 611 Heinz 3402 FL 47 Mountain Fresh | No effects | Roots Crown Stem | USA | [134] |
| *Fusarium oxysporum* *Fusarium fujikuroi* *Neocosmospora solani* * | Momotaro | No effects | Stem | Japan | [135] |
| *Ochroconis humicola* * | Gohobi | Improved plant growth with organic nitrogen sources | Root | Japan | [125] |
| *Albifimbria verrucaria* * *Fusarium* spp. *Setophoma terrestris* *Trichoderma* spp. | Heinz 1706 Moneymaker | No effects | Root | Northern Italy | [133] |

**Table 3.** *Cont.*

| Fungal Species | Tomato Cultivar | Main Results | Location of EF in Plant Tissues | Country | Ref. |
|---|---|---|---|---|---|
| *Sarocladium implicatum* * | Lichun | Biological control suppressed *M. incognita* egg hatching and population, when inoculated to soil | Root | China | [131] |
| *Alternaria* spp.<br>*Aspergillus fumigatus*<br>*Aspergillus nidulans*<br>*Chaetomium globosum*<br>*Coniothyrium aleuritis*<br>*Fusarium chlamydosporum*<br>*Fusarium oxysporum*<br>*Fusarium proliferatum*<br>*Fusarium* sp.<br>*Hypoxylon* sp.<br>*Leptosphaerulina chartarum*<br>*Meyerozyma guilliermondii* *<br>*Neocosmospora solani* *<br>*Nigrospora* sp.<br>*Penicillium helicum* *<br>*Penicillium ochrochloron*<br>*Penicillium simplicissimum* *<br>*Periconia macrospinosa*<br>*Pleosporales* sp.<br>*Rhinocladiella* sp.<br>*Trichoderma atroviride*<br>*Trichoderma spirale* | Big Beef | Plant growth promotion and enhanced fruit weight | Root<br>Shoot<br>Seed | USA | [132] |

* scientific names are different from those present in the articles cited due to taxonomic updates to the name presently use.

## 5. Perspectives on EF Applications to Tomato

EF are ubiquitous microorganisms in the natural and agricultural environment able to colonize plants internally.

In 1994, Dreyfuss and Chapela estimated that the global fungal diversity amounts to 1.5 million species, and based on their estimates, endophytic fungi alone could account for up to 1.3 million species [136]. This perspective on EF diversity was substantiated by subsequent studies of novel plant species, in particular, a study of the fungus:plant ratio in the tropical regions, confirmed that the number of 1.3 million endophytic fungi on the planet was a good assessment [137]. Recently, Hawksworth and Lücking revised the appraisal on global fungal diversity, concluding that the above-mentioned value was too conservative, and the actual range of fungal species should be considered at 2.2–3.8 million species [138].

Although the category of EF is gaining interest in the scientific community, due to their potentially beneficial applications, the studies conducted to date on this topic are still relatively limited and require further investigations.EF can be an extraordinary source of BCAs, PGP, and bioactive molecules, that can provide multiple positive effects to crops, which make them suitable components of biostimulants and biopesticides for use in agriculture [19,26,28]. Most endophytes are considered non-pathogenic, but not all are capable of producing plant beneficial effects [139]. Moreover, even when colonization occurs and positive effects are evident, the costs to the plant in hosting the endophyte/s have to be taken into account, an aspect that has not been studied extensively and is generally underestimated [136,140–143]. It should be considered that EF constitute a rich biodiversity source requiring a greater understanding of: (1) the mechanisms of action involved, those used by the fungal colonizer and the host plant, for crosstalk and recognition that permit the establishment of the interaction; (2) the ecological, biological, and physiological functions of the EF–plant relationship over time; plus (3) the factors and conditions that determine successful colonization [26,144,145].

Understanding the mechanisms underlying the plant-endophyte association and the subsequent outcomes, or the cause and effect, is fundamental for the advancement of EF know-how for the improvement of crop production. Currently, it is well recognized that the interaction between plant and

endophyte is highly influenced by three factors: the genotype of the plant and its microbiome, the fungal genome, and the environmental conditions in which the association occurs [29]. Two major challenges became apparent during the preparation of this review, that in order to develop a wider use of EF in agriculture it will be necessary: (i) to determine how to select the best endophyte–plant combination and establish a stable long-lasting interaction between this beneficial microbe and the plant host targeted for improvement; and (ii) to prove the effectiveness of this technology outside of the controlled test conditions used to date, moving from the greenhouse to the actual open field environment.

Tomato plays host to a microbial community that is vast and highly variable, depending upon the prevailing environmental conditions and the plant genotype [132,144]. It could be interesting to concentrate investigations on the constitutive fungal endophytes that are native to the tomato plant, as they, by their inherent nature, represent a massive pool of highly "tomato-adapted" fungi. This vast fungal community represents a pool of biodiversity that up to now, has been poorly exploited in the strategies to discover highly adapted beneficial microbes of specific crops of interest. In general, a greater comprehension of the mechanisms that favor, along with those that hinder, the endophytic colonization of plants, is required for wider application of EF in agriculture [144]. For example, determine the environmental conditions known to be key factors for a successful EF–plant interaction [26,132].

The artificial introduction of EF in agri-food crops also needs to be analyzed, to ascertain the possible risks that endophytic fungal colonization may present to the plant and the consumer, such as the introduction of potentially toxic metabolites (i.e., mycotoxins) to the food chain [144]. In this respect, studies should assess both the food safety of the fruits produced by EF-colonized plants, as well as evaluate the environmental effects in terms of the release or bioaccumulation of toxins in the soil or crop residues that may be a risk for the agroecosystem. Furthermore, an analysis of the outcome of the endophytic colonization on the organoleptic qualities of the agrifood products should also be taken into account.

In recent years, a growing number of studies have focused on the introduction of beneficial EF to tomato in order to exploit their biocontrol potential against pests and pathogens, as well as their growth promotion effect (references in this review). It is evident from the findings to date, that the introduction of EF represents a promising field of research and development, to which the consequences could determine a remarkable reduction of chemical use in agriculture. This outcome could be clearly observed in the field of crop protection, where it has already been well documented that the biocontrol activity of EF is able to limit the negative effects of several key tomato insect pests and pathogens, as well as nematodes. Moreover, the tomato plants harboring some EF have demonstrated enhanced tolerance to abiotic stress in the field, plus improvement in nutrient uptake, yield, and nutritional quality of the fruits.

This review reports that several EF species are good versatile BCA, controlling both pests and pathogens as demonstrated in the case of *B. bassiana*, *F. oxysporum*, *N. solani*, and *T. harzianum*, which are amenable candidates as plant beneficial microbes, also considering their additional properties as plant biostimulants. Nonetheless, a few surveyed papers considered the possibility to use EF species as a multi-use biocontrol agent, evaluating the simultaneous biocontrol of both pests and pathogens in tomato. Only recently, Jaber and Ownley underlined that some endophytic and entomopathogenic fungi conferred protection to their host plant not only against insect pests but also plant pathogens, and they proposed their use as dual biocontrol agents in agriculture [29]. Another interesting, potential application that has been poorly explored, is the possibility to use different EF species in a consortium and/or with other beneficial microbes. An example is given by the recent work of Varkey and colleagues which has proved that a consortium of rhizobacteria and fungal endophytes suppress the root-knot nematode in tomato [25]. Thus, the possibility to use EF as multiple biocontrol agents and the development of microbial consortia with synergistic beneficial effects on plant performance appears to be an interesting frontier that opens promising fields of research that deserve deeper investigations to better exploit the entire range of EF potential.

## 6. Conclusions

In this review, we summarized the results obtained so far with the artificial introduction of EF in tomato and the subsequent beneficial effects that were observed. The main benefits to tomato plants are attributable to the biocontrol of several insect pests and plant pathogens, as well as their ability to improve plant performance. A focus on naturally occurring, constitutive EF of tomato was also undertaken, aimed at emphasizing their possible role in the selection of new beneficial strains for future use in tomato crop improvement. Moreover, an overview was conducted on the methods of introduction and detection of EF in tomato, providing a clear synthesis of the techniques used, that could be a practical guide to other researchers approaching this interesting field of research. The potential applications of endophytic fungi in horticultural production provide many advantages to the agroecosystem in terms of reducing chemical use and establishing a biological equilibrium necessary for the establishment of sustainable agriculture.

**Author Contributions:** Conceptualization, M.S., M.R., and S.L.W.; literature investigation M.S., M.R., G.d., and L.G.; writing—original draft preparation, M.S. and M.R.; writing—review and editing, S.L.W., project administration, S.L.W.; funding acquisition, S.L.W. All authors have read and agreed to the published version of the manuscript.

**Funding:** This research was funded by the following projects: European Union Horizon 2020 Research and Innovation Program, ECOSTACK (grant agreement no. 773554); fundings and a Ph.D. bursary to M.R. in PRIN 2017 [grant number PROSPECT 2017JLN833].

**Acknowledgments:** Thanks are due to Nadia Lombardi of the University of Naples, Federico II, for her editorial assistance.

**Conflicts of Interest:** The authors declare no conflict of interest.

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
