# Peer review of "Endophytic Fungi of Tomato and Their Potential Applications for Crop Improvement"

_agriculture, doi:10.3390/agriculture10120587_

Round 1

Reviewer 1 Report

Opinion of manuscript – review  entitled “Endophytic Fungi of Tomato and Their Potential Applications for Crop Improvement.

According to the title Authors of manuscript consider the effects and methods of introduction and detection of epiphytic fungi (EF) on tomato plants and indicate the main findings that regard pest and pathogen control, in view of improving plant performance. Moreover, they have discussed on naturally occurring constitutive endophytes present in tomato. They gathered the pertinent literature, and synthesized it in the aim of evaluating the possible role of beneficial EF useful for tomato crop improvement.  Moreover, authors provide an overview of the methods for their isolation and application, discuss the risk of EF application, and indicate the directions of new research on these fungi. This review is useful especially for young scientists, who should pay special attention not only to the amount of yield and profit, but primarily to the possibility of using plant protection products that will be safe for humans and animals. The latest research shows that epiphytic fungi can play such a role. This manuscript is well prepared and is a good source of literature in this field. 

I have the following minor adjustments to propose:

Line 67: adaptation

Line 72: add comma: …growth, as well as....

Line 79 and throughout the paper: Latin names to be written in italics

Line 111: the final part of the sentence (which resulted less productive when feeding on the host plant.) is incorrect, please clarify

Line 137: the sentence ‘Entomopathogenic fungi are a large group of fungi which is traditionally known as insect pathogens.’ Is repetitive, and should be deleted

Line 167: this is the first mention of the species Fusarium solani. The recent nomenclatural change to Neocosmospora solani should be mentioned here, and the species should be referred to as N. solani in the rest of the manuscipt

Table 1, last line to the right: delete (Tucci, 2011)

Lines 299, 301: lycopersici

Line 302: ...symptoms consisting of...

Line 310: delete ‘in Kenya’

Lines 313-314: change ‘Fusarium oxysporum and Neocosmospora solani (formerly known as Fusarium solani)’ to ‘F. oxysporum and N. solani’

Table 3: the species names F. oxysporum and F. fujikuroi should be written in extenso

Line 418: provide

Author Response

Point to point revisions, indicating where changes were made:

  • Line 69: “adaption” was amended with “adaptation”.
  • Line 75: a comma was added as requested.
  • Line 84: the scientific names were formatted in italics.
  • Line 114: the final part of the sentence was deleted while the first part was united with the previous sentence to be more concise and clear. The detailed explanation of how endophytic fungi affect insects is well described in the following sentence.
  • Line 139: the sentence was delated.
  • Line 167: “Fusarium solani” was amended with “Necosmospora solani (formerly Fusarium solani)”.
  • Line 183: “Necosmospora solani (formerly known as Fusarium solani)” was amended with “ solani”.
  • Table 1: last line to the right: “(Tucci, 2011)” was delated.
  • Lines 304 and 305: “licopersici” was amended with “lycopersici”.
  • Line 306: “on consisted” was amended with “consisting of” and then removed/ the entire sentence was revised.
  • Line 313: “in Kenya” was delated
  • Lines 316: “Fusarium oxysporum and Necosmospora solani (formerly known as Fusarium solani)” was edited in “ oxysporum and N. solani”.
  • Table 3: the species names oxysporum and F. fujikuroi were written in extenso.
  • Line 432: “provides” was amended with “provide” provide.

Reviewer 2 Report

The manuscript reviewed the research progress of endophyte application on tomato. The manuscript provided comprehensive and up to date information with well organized materials. I have no disagreement in science with the fact and opinions in the manuscript. The manuscript is written in fluent English. There are still some minor problems on language, and it would be better to be checked with a language editor. 

Comments in text:

L 56, the expression of this sentence is deficient and not specific. “In recent decades” should refer to research progress in recent decades. Probably better to modify the sentence, for example, “Based on studies in recent decades, it has......” or “In recent decades, scientific evidence showed it apparent that......”.

L 106, it seems an adverbial is acting as a subject. Please remove “Among”. For other minor grammar errors, probably better to write “The beneficial effects carried out by EF include pests and pathogens biocontrol (as biocontrol agent, BCA),”.

L 123 – 127 is repeating L 106 – 108.

L 133, author should decide where to introduce the term “Biological Control Agents” for the first time, at L133 or L107, since the abbreviation “BCA” should follow the full name in parenthesis only once. Afterwards the abbreviation should not be presented in parenthesis again.

L 141, “suggests their ability of endophytic colonization on a wide range of”

L 149, Piriformospora indica and P. williamsii have been transferred to genus Serendipita in 2016 (M. Weiss, F. Waller, Zuccaro & Selosse, New Phytol. 211(1): 20-40)

L 205, no need to capitalize “Fungi”

L 343, I would suggest to carefully use the number 1 million species of endophytes. I didn’t find the original literature in citation 26. And the most recent publication referring to global fungal diversity estimation is: Hawksworth and Lücking 2017. Fungal Diversity Revisited: 2.2 to 3.8 Million Species. In Heitman et al (ed), The Fungal Kingdom. ASM Press, Washington, DC. doi: 10.1128/microbiolspec.FUNK-0052-2016. Hawksworth and Lücking didn’t give a number for endophytes in it.

Author Response

Point to point revisions, indicating where changes were made:

  • Line 57: the expression “In the recent decades” was replaced with “In recent decades, scientific evidence has demonstrated that...”.
  • Line 110: “Among” and “BCA” were removed and the period was edited.
  • Line 127-130: the whole sentence was edited to avoid the repetition.
  • Line 135: the term “Biological Control Agents” and its abbreviation was introduced in this sentence.
  • Line 142: “skill” was amended with “ability”.
  • Line 149: “Piriformospora indica” was edited in “Serendipita (formerly Piriformospora)”/ the entire paragraph was edited.
  • Line 204: the word “Fungi” with capital letter was not found in this sentence nor anywhere in the whole manuscript, please let us know if you mean something else.
  • Line 350: the sentence was revised and the original reference was added: Dreyfuss, M.M.; Chapela, I.H. Potential of fungi in the discovery of novel, low-molecular weight pharmaceuticals. Biotechnology. 1994, 26, 49-80. doi:10.1016/b978-0-7506-9003-4.50009-5
    Moreover, an additional and more recent reference in which the estimate by Dreyfuss and Chapela was commented and substantiated was added: Hawksworth, D.L. The magnitude of fungal diversity: the 1.5 million species estimate revisited. Mycol Res, 2001,105,1422–1432, http://dx.doi.org/10.1017/S0953756201004725.
    Finally, the reference that you suggested about the more recent estimate of the global fungal diversity, that is from the same author of the latter reference we cited, was added.